# Need a Small Specialized Language Model? Plan Early!

## Abstract

Large language models are versatile tools but are not suitable for small inference budgets. Small models have more efficient inference, but their lower capacity means that their performance can be good only if one limits their scope to a specialized domain. This paper explores how to get good specialized small language models using a large, generic, pretraining set and a limited amount of specialized data. We consider two scenarios, depending on whether (i) one can afford pretraining a model for each specialization task, or (ii) one wants to cheaply adapt a single pretrained model for each task. In the first scenario, we propose an effective solution based on importance sampling: we resample the pretraining set to imitate the specialization data and train a small model on it. In the second scenario, we propose a novel architecture, projected networks (PN). PN is a large network whose parameters can be linearly projected into a small network for specialization. For both scenarios, we demonstrate the empirical effectiveness of our solutions across various domains, training set sizes, and training budgets.

## 1 Introduction

Large Language Models (LLMs) have emerged as a generic tool to address a wide range of language tasks (Brown et al., 2020; Bommasani et al., 2022). This generality requires a large, diverse, generic training set. This rich set ensures that the model fits many subdomains close to the tasks it will eventually address. Model generality is particularly impactful for tasks where the cost of collecting a representative training set cannot be justified. However, LLMs inference is costly because of their large number of parameters, required to fit a large training set. This cost restricts LLMs to high-value applications. Efficient inference is an active research area that follows multiple routes like model distillation (Hsieh et al., 2023), quantization (Dettmers & Zettlemoyer, 2023), pruning (Ma et al., 2023) or hardware optimization (Aminabadi et al., 2022). However, reducing model size is the most direct solution for applications under tight inference constraints, and it can be combined with all the aforementioned techniques to further reduce inference costs. With that inference constraint in mind, we focus on training Small Language Models (SLMs).

A SLM cannot fit a generic training set as well as an LLM (Vapnik, 1995; Bishop & Bishop, 2023). It is hence necessary to forgo the generality of the model to devote its limited capacity to a targeted specialization domain. While a large specialized training set for the application at hand would be ideal for training, such a set is costly and usually justified only for high-value applications. Many applications, therefore, have to face both a limited inference budget and a limited in-domain training set size. For instance, some applications cannot afford to collect more than a few million tokens worth of training data (1m tokens $\simeq$ 10 books). Such applications, with low inference and low data collection budgets, are facing a challenging problem.

This work tackles precisely this problem. We aim to answer the following question: **How can we get a *specialized* small language model when little specialization data is available?** To answer this question, we distinguish two scenarios of practical importance, each leading to different recommendations: (i) when one needs a single small model for **one specialized domain**, a large training budget can be considered for that domain and one can pretrain a model specifically targeted to this domain; on the other hand, (ii) when one needs one model per domain for **many specialized domains**, pretraining one model per domain quickly becomes too expensive, and it is interesting to share the training cost across domains.

Figure 1: Practical recommendations for training LMs that fit a predefined computational budget.

For both scenarios, there are natural baselines that come to mind. The first baseline consists of pretraining a small model using the large generic pretraining set and then fine-tuning it with the little available specialization data. Another widely used method is distillation (Hinton et al., 2015): one pretrains a large teacher and a small student model on the pretraining set, fine-tunes the large model on the specialization set, and then fine-tunes the small model with the distillation loss from the fine-tuned large model.

While these two baselines are widely used in practice, for both scenarios, we propose improved pretraining strategies and demonstrate their superiority. The cornerstone of our improved strategies is a clustering of the pretraining set, which allows us to sample data points from each cluster of the pretraining set and pretrain on a mixture of these clusters with the flexibility to choose the weight of each cluster. When one targets a single domain (i), we find that an effective method is pretraining a small model over generic data with *importance sampling* (Owen, 2013). We pretrain on a mixture of the clusters where the weights are chosen to mimic the targeted domain. We show that, even though this emphasis relies on information from the specialization data, such a pretrained model still benefits from fine-tuning over the specialization data and outperforms the previous baselines. When one targets multiple domains (ii), we propose a novel architecture for pretraining, *the projected network* (PN). This model is a large model trained on generic data in which some parameters are tied to a cluster. Crucially, its parameters can be linearly projected into different small models — one per cluster — prior to fine-tuning. When one is given a new specialization domain, we show that it is effective to select a single projection and fine-tune the corresponding small model. In this paradigm, training the generic high-capacity model is costly, but its specialization phase (projection onto an SLM and SLM fine tuning) is not. This is ideal when many specialized models are needed.

We evaluate each method on several domains, pretraining and specialization budgets as well as specialized dataset sizes. Compared to the baselines, both importance sampling and projected networks bring a strong improvement, on all considered domains, specialized dataset sizes and training budgets. We show that importance sampling results in better specialized-perplexity than projected networks. However, importance sampling incurs a high pretraining cost when targeting multiple domains since pretraining is not shared across domains. As a result, projected networks are beneficial in the second scenario, for applications requiring many specialized models.

**Contributions** In Section 2, we present classical pretraining strategies as well as our novel methods based on a clustering of the pretraining dataset. We present our cluster-based importance sampling strategy. We introduce Projected Networks, a novel application for hard mixture models and hypernetworks, and explain how this provides multiple models that can be quickly instantiated during the specialization phase. In Section 3, we propose a methodology to evaluate different strategies to train a specialized SLM and identifies four important variables: the generic training budget (for training before the target domain is known), the specialization budget (for training after the target domain is known), the inference budget, and the in-domain training set size. In Section 4, we provide a comprehensive empirical study by reporting experiments on 9 domains, 3 specialized training set sizes with various training budgets. We show that, as expected, fine-tuning is essential. Surprisingly, we highlight that distillation from a large model, albeit popular, brings little improvement when accounting for the overall pretraining budget. We show that our cluster-based importance sampling method is very effective for training specialized models in the large specialization budget case. Finally, we find that Projected Networks lead to the best models in the small specialization budget case. Figure 1 summarizes our practical recommendations.

## 2 METHODS

We consider different strategies to leverage a large, generic pretraining set under inference constraints.

## 2.1 Generic and Specialization Datasets

We have access to a large, generic pretraining set, which is typically obtained as the output of a web crawl. This set is *large*: it has enough samples to train a model on it or on a subset of it without risks of overfitting. This set is *generic*: it covers a wide variety of topics and concepts. We train models on this set using the next-token prediction loss.

We also have access to a small specialization set, which can be, for instance, the internal documentation of a company, some emails, or a set of scientific articles. This set is *small*: one cannot train a good model on it without overfitting. This set is *specialized*: it only covers specific topics. Once again, we measure the quality of models on this set with the next-token prediction loss and perplexity.

## 2.2 Baselines: Small Models, Fine-Tuning & Distillation

We target a model with a good specialized perplexity under inference constraints. We abstract the inference constraints as a limit on the model size. We call **Small Language Model (SLM)** a transformer of the maximal possible size allowed by the inference constraint.

We consider 3 variants of SLMs. We first consider an SLM model trained only on the generic pretraining data. This model can be effective if the generic and specialization data are close. We then consider an SLM trained only on the specialization data . This model can be effective if the specialization training set is large enough but will quickly overfit when the specialization set is small. Finally, we consider a model pretrained on the generic data and fine-tuned on the specialization data. The amount of fine-tuning can be adjusted via early stopping. This adjusts a trade-off between the proximity to the generic distribution and overfitting to the specialization set. Training and fine-tuning are done by minimizing the next-token prediction loss on the corresponding datasets.

Since our capacity constraint is motivated by inference efficiency, we can consider larger models at training time and rely on distillation (Hinton et al., 2015) to satisfy the inference requirements. Specifically, we consider a large language model (LLM) that has a bigger size than the inference constraint allows. We pretrain it on the generic pretraining set, and then fine-tune it on the specialization set, achieving a lower perplexity on the specialization set than the pretrained-then-fine-tuned SLM thanks to its larger size. A pretrained SLM student is then fine-tuned to a mixture of the specialization data and next-word distributions from the teacher. We call this model **SLM-d**.

This type of distillation is often called *Response-based Knowledge distillation* in the literature (Gou et al., 2021, Sec 2.1) and is arguably the most widely used type of distillation. We do not consider using the LLM as a data augmentation tool (Feng et al., 2021), to produce synthetic data (Tang et al., 2019; West et al., 2021) resembling the specialization data, and then use them to train or fine-tune the small model. Indeed, this requires training an LLM, which is time-consuming, and in our setup, this leads to more variance than response-based knowledge distillation and overall poorer performances (Menon et al., 2021).

## 2.3 Clustering of the Pretraining Data

The subsequent strategies rely on a clustering of the generic pretraining set. Clustering associates each training example with a discrete latent variable that we use to efficiently estimate importance sampling weights with a conditional independence assumption (Section 2.4). It also allows us to condition the projection in projected networks (Section 2.5).

To cluster the data, we embed each document in the generic set as a vector using Sentence BERT (Reimers & Gurevych, 2019). The documents longer than the maximum context we consider (1,024) are broken into nonoverlapping windows. Then, we use the k-means algorithm to cluster the generic set into $k$ clusters. We can then query samples from each cluster. This also defines a cluster assignment function $\text{assign}(x)$ for any sample $x$.

## 2.4 Cluster-Based Importance Sampling

Importance Sampling (IS) is a well established method (Owen, 2013) that enables estimating the expectation of a variable (the next-token prediction loss in our case) on a distribution (the specialization distribution), while sampling from a different distribution (the generic distribution). In practice, IS

needs to estimate importance weights from data. We leverage the pre-training set clustering proposed above to make the importance weight estimation easy, by assuming our data are sampled from a mixture over clusters. Letting $\ell$ the individual loss function, the specialization loss is

$$\mathcal{L}(D_{\text{spec}}; \theta) := \mathbb{E}_{x \sim D_{\text{spec}}}[\ell(x; \theta)] = \sum_x \ell(x; \theta) P(x|D_{\text{spec}}).$$

We introduce a latent cluster membership variable $c$, and make an independence assumption $P(x|c, D_{\text{spec}}) = P(x|c)$, which gives

$$\mathcal{L}(D_{\text{spec}}; \theta) = \sum_x \sum_c \ell(x; \theta) P(x|c, D_{\text{spec}}) P(c|D_{\text{spec}}) = \sum_x \sum_c \ell(x; \theta) P(x|c) P(c|D_{\text{spec}}).$$

We then apply importance sampling,

$$\mathcal{L}(D_{\text{spec}}; \theta) = \sum_x \sum_c \ell(x; \theta) P(x|c) \frac{P(c|D_{\text{spec}})}{P(c|D_{\text{generic}})} P(c|D_{\text{generic}}) = \mathbb{E}_{x \sim D_{\text{generic}}}[w(x)\ell(x; \theta)]$$

with the importance weight $w(x) = \frac{P(c(x)|D_{\text{spec}})}{P(c(x)|D_{\text{generic}})}$ and $c(x)$ denotes the single cluster $c$ such that $P(x|c) > 0$. The importance weights can, therefore, be estimated as the ratio between the cluster frequencies in the generic and specialization training set. In other words, we compute the histograms $h_c = \frac{\#\{x \in D_{\text{spec}} \text{ s.t. assign}(x)=c\}}{\#D_{\text{spec}}}$, and then train the model by sampling from a mixture of the clusters with frequency $h_c$. The number of cluster $k$ is a trade-off between large $k$ (unreliable cluster frequency estimates for the specialization set, risk of overfitting to that set) and small $k$ (stronger independence assumption when the clusters are large). The small models trained with importance sampling are called **SLM-is**.

## 2.5 ASYMMETRIC MODELS: PROJECTED NETWORKS AND HARD MIXTURES

Our inference constraints require that the final specialized model is a low-capacity SLM. Prior to fine-tuning on the specialization data, the capacity limit does not apply to generic pretraining. We devise a pretraining strategy to take advantage of this asymmetry. At pretraining time, we train a network with many parameters, but each example only interacts with a projection of the parameters onto an SLM. Like distillation, this strategy trains a model with many parameters, but unlike distillation, all model evaluations during training are already constrained to operate within the size limits.

**Projected network** Our Projected Network (PN), **SLM-pn**, trains jointly a collection of small models or *experts*, $\{\text{SLM-pn}_i\}_{i=1}^k$; there is one expert per cluster $i$. Each expert is instantiated via its specific linear projection of the large parameters; see Figure 2. We train a PN network with many parameters during generic pretraining. Once the specialized training data are available, specialized fine-tuning starts from one of the experts. Different strategies for expert selection are discussed in our experiments.

The PN model adds parameters to the linear layers of a model. It is configured via 3 hyper-parameters $h, k, m$. $h$ is a multiplicative factor increasing the overall number of parameters while $k$ controls the number of experts / clusters. Finally, $m$ controls the number of parameters specifically allocated to each expert. For each SLM-pn$_i$, the parameter matrix $W^{(l,i)} \in \mathbb{R}^{d \times d'}$ of a layer $l$ is computed via a linear projection,

$$W_{a,b}^{(l,i)} = \sum_{q=1}^m E_{i,q} \sum_{r=1}^h M_{q,r}^{(l)} T_{a,b,r}^{(1,l)} \text{ for } a = 1 \ldots d \text{ and } b = 1 \ldots d'$$

where $E_i \in \mathbb{R}^m$ is an expert-specific vector, $M^{(l)} \in \mathbb{R}^{m \times h}$ is a layer-specific matrix and $T^{(1,l)} \in \mathbb{R}^{d \times d' \times h}$ is a tensor that stores most of the parameters. In our experiments, our SLM-pn experts are transformers and we only apply the PN decomposition to the feed-forward layers (i.e. multi-layer perceptron, MLP) which hold most of the model parameters. The other parameters are shared.

The PN can separately set the overall network size via $h$ and the number of distinct experts via $k$. We train one expert per cluster, using the clustering from Section 2.3. We associate each training example $x$ with a cluster variable $c(x) = 1 \ldots k$ and its loss on $x$ is computed with SLM-pn$_{c(x)}$. Training optimizes all experts jointly by minimizing the expected loss on the generic set.

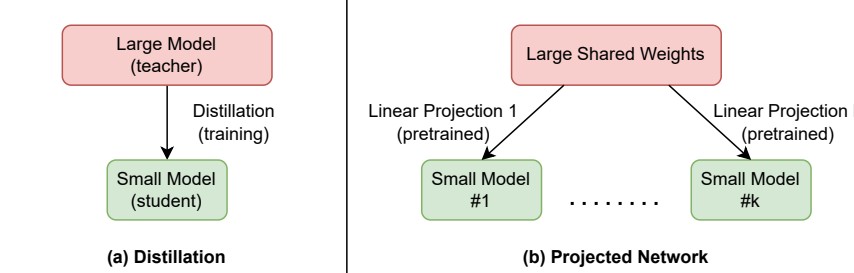

Figure 2: Projected networks (right) unlike distillation (left) instantiate small models in closed-form.

For specialization, we select one expert SLM-pn$_i$ and fine-tune it on the specialization dataset. Among the different strategies we evaluate for expert selection, we find that picking the pretrained expert corresponding to the most frequent cluster in the specialization dataset is an effective strategy.

**Hard Mixture of Experts** As a simple alternative to the previous architecture, we consider **SLM-mix**, a hard mixture of expert (Gross et al., 2017). We pretrain one SLM, SLM-mix$_i$, on each cluster $i$, independently. The pretraining cost and overall number of parameters of this method are high as both scale linearly with the number of clusters. The hard mixture can be compared to a special case of a PN network in which $h = k$, $E_i = \delta_i \in \mathbb{R}^k$ and $M^{(l)} = $ I. In that case, all weight matrices rely on independent slices of the parameters. Unlike PN, the hard mixture does not allow one to set the number of experts $k$ and the capacity multiplier $h$ independently. The expert parameters are not shared and learning cannot leverage synergies between similar clusters. On the other hand, the learning is embarrassingly parallel since each expert pretraining is independent of the other experts. Like for PN, specialization can be performed inexpensively by fine-tuning a single expert. Despite these conceptual differences, our experiments reveal benefits in both methods.

# 3 EXPERIMENTAL SETUP

## 3.1 METHODOLOGY

With inference cost and specialization training data constraints, we study the alternative training methods at various training costs. We report training costs and which part of the cost can be shared across multiple domains. We consider 4 important metrics:

**Generic training cost**: the cost of the training phase that can be performed before the specialization data are available, on a generic training set. This cost is often called pretraining. It is domain-independent and can be shared across multiple specializations, e.g., via later fine-tuning. Although not mandatory, the generic training data are essential when specialization data are limited.

**Specialization training cost**: the cost of the training performed once the specialization data are available. This cost is not shared across different specializations.

**Inference cost**: the cost of running inference on a specialized model. Low inference cost allows wider, cheaper model deployment.

**Size of the specialization training set**: it varies across applications and influences pretraining and specialization choices.

Taking the inference cost and the specialization data size as hard constraints, we study the operating curves resulting from varying the generic and specialization training costs. We measure training cost (pretraining and specialization) in hours of graphic processor compute time (GPUh) on the same hardware (Nvidia-A100). We consider pretraining costs ranging from 10 to 650 GPUh and specialization costs ranging from 0.3 to 120 GPUh.

We evaluate language modeling with perplexity, using 20k held-out documents per dataset. We focus solely on language modeling; evaluating the models on downstream tasks (e.g. question answering, sentiment analysis, translation, etc) is beyond the scope of the paper. However, our conclusions could extend to downstream tasks as perplexity and downstream performance are often correlated Gonen

Table 1: Number of parameters (millions) for generic pretraining and inference. SLM-mix and SLM-pn are large models during pretraining but small at inference.

| Model | Num. parameters (M) | |
|---|---|---|
| | Pretrain | Inference |
| SLM | 126 | 126 |
| SLM-mix | 2,016 | 126 |
| SLM-pn | 1,422 | 126 |
| LLM | 771 | 771 |

Table 2: Model throughput (GPU hours per 1B training tokens). Inference of SLM is $\sim 4x$ faster than LLM.

| Model | Training | | Inference |
|---|---|---|---|
| | Generic | Specializ. | |
| SLM | 2.2 | 2.2 | 0.61 |
| SLM-mix | 2.2 | 2.2 | 0.61 |
| SLM-pn | 3.6 | 2.2 | 0.61 |
| SLM-is | N/A | 2.2 | 0.61 |
| LLM | 7.7 | 7.7 | 2.54 |

Table 3: Train cost upper limits for pretraining and specialization (GPUh). Specialization is inexpensive except for SLM-is, SLM-d.

| Model | Pretraining | Specialization | | |
|---|---|---|---|---|
| | | 1M | 8M | 64M |
| LLM | 650 | 0.12 | 0.5 | 3.5 |
| SLM | 530 | 0.02 | 0.07 | 0.5 |
| SLM-is | 0 | 130 | 130 | 130 |
| SLM-d | 1,850 | 0.7 | 2.8 | 21 |
| SLM-mix | 650 | 0.02 | 0.07 | 0.5 |
| SLM-pn | 650 | 0.02 | 0.07 | 0.5 |

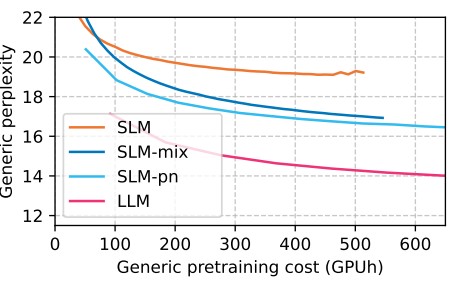

Figure 3: Generic pretrain perplexity on c4.

et al. (2022); Du et al. (2024); Gadre et al. (2024). We report perplexity on generic and specialization data. For the different specialization domains. We report perplexity per domain in Appendix E and present *macro-averaged* results in Section 4. For macro-averaged perplexity, we compute the mean negative log-likelihood per token for each domain, average these results, and compute the exponential. All domains therefore get the same weight, regardless of the number of tokens per held-out set.

## 3.2 DATASETS

Our generic pretraining set is c4, a large filtered dataset of English text derived from common-crawl (Raffel et al., 2020). We tokenize the data with a sentence piece model trained on c4 with a vocabulary size of 32k. We use the clustering method described in Section 2.3 to split this dataset into $k$ clusters. We use $k = 1024$ for SLM-is, and $k = 32$ for SLM-pn and SLM-mix. We investigate the impact of the number of clusters in Appendix D.

We consider specializing to nine diverse domains, extracted from the Pile (Gao et al., 2021): arxiv (science articles), europarl (parliamentary proceedings), freelaw (legal text), gutenberg (old books pusblished before 1919), opensubtitles (theatrical subtitles), openwebtext2 (forum discussions), pubmed-abstracts (medical article abstracts), stackexchange (Q&A mostly about technical topics), wikipedia (encyclopedia articles). We vary the amount of specialization training data available and consider sets of size 1, 8 and 64 million tokens for each domain.

## 3.3 MODELS HYPER-PARAMETERS

Table 1 reports the number of parameters for the pretrained and specialized models. Table 1 illustrates that SLM-pn and SLM-mix (Section 2.5) are as small as SLM for inference after specialization while their overall number of pretrained parameters is larger than LLM. Table 2 reports the throughput of the models. All SLM models have the same specialization throughput while SLM-pn has a lower throughput than SLM, SLM-mix for pretraining. LLM is more expensive in all cases. Table 3 presents the upper limit in training budgets for pretraining and specialization over all settings. Appendix A reports the training hyperparameters.

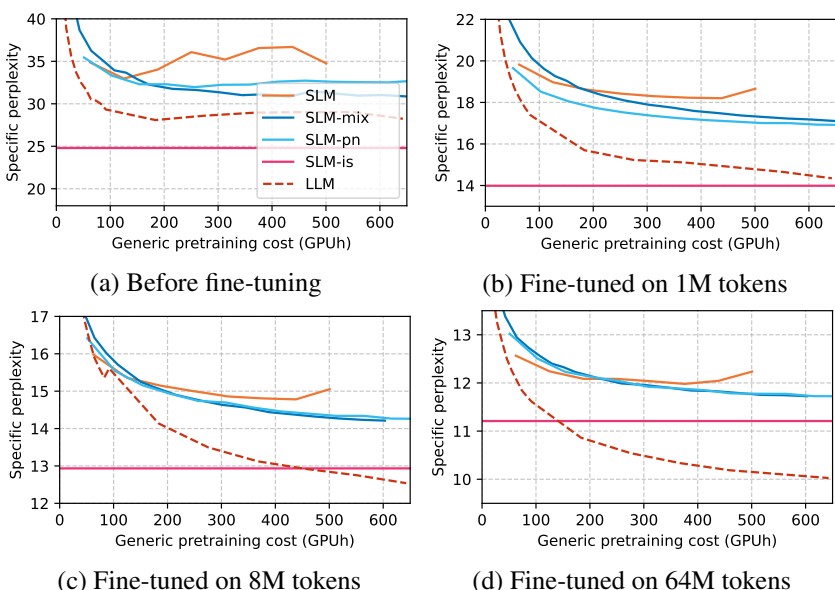

(a) Before fine-tuning        (b) Fine-tuned on 1M tokens

(c) Fine-tuned on 8M tokens       (d) Fine-tuned on 64M tokens

Figure 4: Specialized perplexity on the Pile subsets (average) before and after fine-tuning with different amounts of specialization data. Fine-tuning is necessary to reach good specialized perplexity for all models. We display SML-is as a flat line since it has no generic pre-training phase; all of its cost is in the specialization phase. With 1m specialization tokens, SLM-is competes with the LLM.

## 4 EMPIRICAL EVALUATION

We first report our main results before comparing the different methods. We vary pretraining budgets and report perplexity on the generic pretraining set (c4) for each method in Figure 3. When we consider SLM-pn and SLM-mix, we observe that even if the number of pretrained parameters is larger than LLM, they do not enjoy as good perplexity. However, their perplexity is better than SLM while they are as efficient when tested or fine-tuned on a single cluster.

Generic perplexity (c4) is not our primary goal and we now examine specialized perplexities. Figure 4 (a) reports the results before fine-tuning. Specialized perplexities are much higher than the c4 perplexities, indicating that specialization is necessary. Figure 4 (b) reports the results after fine-tuning several pretrained checkpoints for each method on the 1M token dataset of each domain. Each domain-specific model is evaluated before macro-averaging. Since 1M tokens is a small set, fine-tuning relies on a small learning rate and early stopping (base learning rate divided by 3, stopping when validation loss stops improving, which is always less than 2k fine-tuning steps on one GPU when validation loss stops improving). Fine-tuning is highly beneficial for all methods and results in significantly improved perplexity. We also remark that pre-fine-tuning perplexity on the Pile is not necessarily a good indicator of post-fine-tuning perplexity: e.g. the SLM checkpoints ordering is very different on the two curves, the ordering between SLM-mix and SLM-pn also changes during fine-tuning.

We also consider fine-tuning on 8 and 64 million tokens for each domain, see Figure 4 (c) and (d). More data allows us to train slightly longer and keep the base learning rate without overfitting. We stop at most after 4k steps and 30k steps for the 8M and 64M cases respectively. We observe that the benefit of a good starting point provided by SLM-pn and SLM-mix (compared to SLM) erodes as the domain training set size increases.

We report the perplexity of SLM-is as a constant line. This method has no generic pretraining as its training starts only once the domain data are available; bearing all the training cost in the specialization phase. SML-is is the best method with a small inference model in terms of post-specialization perplexity. Interestingly, it even outperforms the much larger model when specialization data are scarce (ie the 1M tokens case), for a fraction of the overall training cost (¡130 GPUh).

Table 4: Perplexity on the Pile (average) for small and large LMs ($< 650$GPUh of pretraining). SLM-nopt is an SLM that is trained directly on the specialization set, without pre-training.

Table 5: Selecting the best expert for SLM-mix. Average specialized perplexity fine-tuned over 1M tokens, 64 experts, after 700k pretrain steps ($\sim 600$ GPUh). Post-fine tuning selection performs slightly better but is more costly.

| Model | Pretrained | Specialized | | |
|---|---|---|---|---|
| | | 1M | 8M | 64M |
| SLM | 33.0 | 18.2 | 14.8 | 12.0 |
| SLM-nopt | N/A | 227.1 | 45.6 | 17.6 |
| LLM | 28.1 | 14.4 | 12.5 | 10.0 |

| Method | $P_{\text{erplexity}}$ | Fine-tune cost |
|---|---|---|
| Most frequent cluster | 17.32 | 1x |
| Best pretrained | 17.05 | 1x |
| Best fine-tuned | 16.98 | 64x |

## 4.1 BASELINES: FINE-TUNING, DISTILLATION

Table 4 compares the perplexity on the Pile subsets for the baseline transformer models. Pretraining and fine-tuning are both necessary to achieve good perplexity on our specialization sets. Without pre-training, a lot of specialization data (64M tokens per domain) is needed to get acceptable performance. For both large and small models, there is a large gap in perplexity before and after finetuning, making it clear that finetuning even on 1M in-domain tokens can result in a significant boost in performance. Finally, as expected, the LLM results also illustrate that, for large inference and pretraining budgets, it is beneficial to train large models on the pretraining set (c4). We investigate the effect of applying parameter efficient fine-tuning in Appendix F; this does not change the previous conclusions.

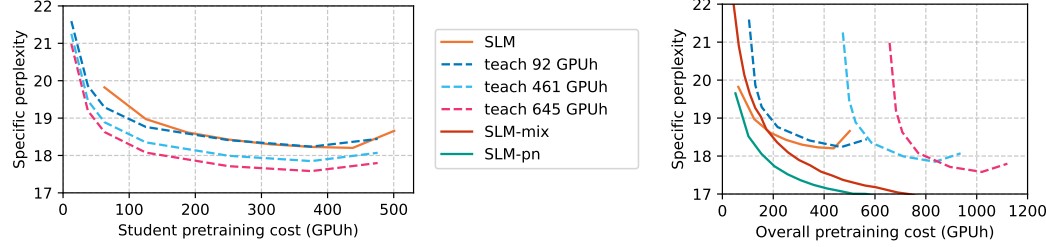

Figure 5: Distillation results (dashed lines) on the 1M token specialization set for various teacher pretraining budgets. On the left we show perplexity with respect to the student pretraining cost only and on the right with respect to the overall pretraining cost. The cost of distillation is high when compared to its benefit compared to SLM-mix, SLM-pn.

Our distillation process takes a pretrained teacher (LLM) and a pretrained student (SLM). We fine-tune the teacher on the specialization set and we use the fine-tuned teacher to supervise the student on the same set. In this process, the generic pretraining cost sums two terms: teacher and student pretraining. Figure 5 (left) reports SLM-d perplexities with each curve corresponding to a different amount of teacher pretraining and has the student pretraining as the x-axis. It shows that for settings over 276 GPUh of teacher pretraining (300k steps), the student model SLM-d is significantly better than vanilla SLM at the same level of student pretraining. This plot demonstrates the benefit of a good teacher over an SLM trained only over the specialization set. Figure 5 (right) shows SLM-pn and SLM-mix achieve a better specialized perplexity than SLM-d when comparing overall pretraining costs, which accounts for the cost of teacher training. Even without counting the cost of teacher training cost, the benefit of SLM-d quickly vanishes compared to SLM-pn and SLM-mix.

## 4.2 IMPORTANCE SAMPLING

Our importance sampling strategy resamples the generic set (c4) such that its cluster histogram matches the cluster histogram from the specialization set (Pile subset). This paradigm requires a different resampled generic dataset for each specialization task. This makes SLM-is costly when addressing many tasks. For a model, the total cost of specialization over $N$ tasks is

$$C_{\text{total}}(N) = C_{\text{generic}} + C_{\text{specialization}} \times N. \tag{1}$$

For methods like PN, most of the cost is $C_{\text{generic}}$ and the main parameter to vary the total cost is the number of generic pretraining steps. For the importance sampling method, $C_{\text{generic}} = 0$ and the main parameter to vary the total cost is the number of steps performed when training on the importance sampled pretraining set, which is part of $C_{\text{specialization}}$.

We vary the total cost for SLM-pn and SLM-is when hypothetically addressing 1, 7 and 50 tasks by scaling the x-axis following Equation 1. Figure 6 shows that SLM-is becomes less interesting when the number of tasks increases. The specialization cost of fine-tuning for SLM-pn, which increases linearly with the number of tasks, can be ignored as it takes $\sim 1$GPU minute.

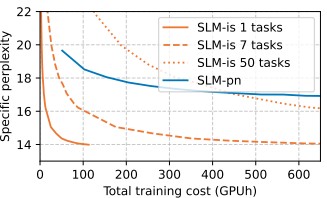 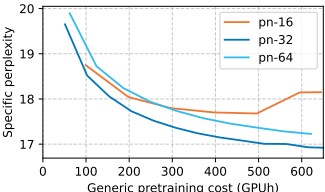 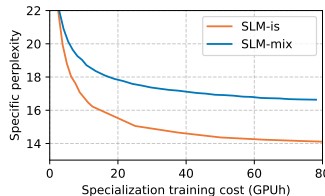

Figure 6: Specialized perplexity for SLM-pn vs SLM-is after fine-tuning on 1M. SLM-is cost increases linearly with number of tasks.

Figure 7: Specialized perplexity for PN with different number of experts after fine-tuning on 1M tokens.

Figure 8: Specialized perplexity vs specialization cost after fine-tuning on 1M tokens, when one is only training SLM-mix on the most frequent domain cluster.

### 4.3 Asymmetric models: Hard Mixture of Experts and Projected Networks

The **projected-networks** allow one to select the overall number of parameters of the network while keeping the size of the inference model constant. Unlike the hard mixture of experts, varying capacity does not require changing the number of clusters either. The PN capacity is therefore a trade-off between generic pretraining cost and specialized accuracy. Figure 7 shows perplexity on the Pile subsets after fine-tuning on 1M tokens as a function of pretraining cost. While more experts always perform better per iteration, 32 experts achieve a better cost/perplexity trade-off.

The **hard mixture of experts** relies on the generic dataset split in clusters, see Section 2.3, and its number of experts corresponds to the number of clusters. For specialization, we fine-tune a single expert. The results presented above, e.g. Figure 4, use the expert corresponding to the most frequent cluster in the specialization data. Alternately, we also consider selecting the expert which has the lowest loss on the specialization set before fine-tuning, which involves evaluating each expert. As a third more costly option, we fine-tune all experts and pick the best one a posteriori. Table 5 reports this result when fine-tuning on 1M tokens with 64 experts. The results of the different strategies are close, $\pm 0.3$ PPL, and the most costly option of fine-tuning all experts performs slightly better.

As a final observation on SLM-mix, the strategy of fine-tuning only the expert corresponding to the most frequent cluster can be very efficient when one targets a single domain. In that case, one can train only the expert for the single cluster of interest. This single-cluster training strategy is however a poorer approximation of the specialization distribution than IS, as shown in Figure 8.

## 5 Related Work

Domain adaptation for language modeling has a long history, predating neural network language models Rosenfeld (2000). This research stemmed from the observation that models trained on large amount of data, even far from the targeted domain were impactful on end applications Brants et al. (2007). After neural language models were introduced Bengio et al. (2000), they were also scaled up to benefit from increasing amount of training data Raffel et al. (2020); Brown et al. (2020); Chowdhery et al. (2022); Touvron et al. (2023). This growth involves a trade-off between training a model from a large dataset (i.e. reducing estimation errors) or a dataset representative of the end application domain (i.e. having a training distribution representative of test condition), both essential to good generalization Vapnik (1995).

Model fine-tuning and multi-task learning have become essential tools in order to both benefit from large generic training data and limited in-domain data Caruana (1993); Collobert et al. (2011); Gururangan et al. (2020). Data curation and selection methods have also been proposed in order to resample generic data with a given application domain in mind Moore & Lewis (2010); Wang et al. (2018); Xie et al. (2023). Most of these methods can be tied to importance sampling Kahn & Harris (1951); Grangier & Iter (2022).

Simultaneously with the growth in large language model size, concerns about model inference cost gave rise to research on efficient inference. Several routes are investigated with this goal, including model distillation Hsieh et al. (2023); FitzGerald et al. (2022), weight quantization Xiao et al. (2023); Dettmers & Zettlemoyer (2023) and pruning Ma et al. (2023); Xia et al. (2023). Mixtures of experts have been investigated as a way to decouple overall model capacity and inference efficiency Shazeer et al. (2017); Du et al. (2022); Clark et al. (2022).

Our asymmetric projected network can be seen as a hyper-network, a type of neural network whose parameters are themselves predicted by a secondary network Ha et al. (2017); Karimi Mahabadi et al. (2021). In our case, the secondary network is a cluster-conditioned linear projection. Gururangan et al. (2023) also propose learning one model per domain of interest and using an ensembling technique at inference time, which requires instantiating multiple models.

## 6    LIMITATIONS

Our experimentation covers multiple domains, training budgets and training set sizes but, at this point, we did not explore multiple sizes for our SLMs. We want to verify in the future if the advantage of PN over distillation extends to different pair of large/small model sizes. Similarly, we studied the impact of the number of clusters on SLM-is, SLM-pn and SLM-mix but all our clustering experiments represent documents with sentence BERT, while different representations might impact our results.

## 7    CONCLUSIONS

This work considers a common double practical constraint for language modeling: the scarcity of in-domain training data and a limited inference budget. We propose to train small, efficient language models and improve their accuracy by rethinking the pretraining process on abundant, generic training data. This paper formalizes the problem and proposes two main contributions. (i) When one can afford pretraining a model per specialization domain, we introduce an importance sampling method based on data clustering. This allows pretraining to focus on data close to the targeted domain. (ii) When one needs to share the cost of pretraining across multiple specialization domains, we propose Projected Networks, a novel architecture that trains a collection of small models jointly. Each model of the collection can be used on its own, for instance, for fine-tuning on a new domain. The empirical benefit of our contributions is shown across multiple domains, training budgets and training set sizes. Our work yields simple recommendations summarized in Figure 1. Another benefit of the projected networks is that they can be specialized without access to pretraining data. For instance, if the specialization data is sensitive, the data owner can cheaply instantiate and fine-tune the model themselves. Our methodology is not specific to language modeling and we plan to extend it to other modalities where inference constraints are also important (e.g. computer vision).

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

## A  Hyper-parameters

All our language model are either instances of SLM or LLM. We rely on the parameters from Table 6. Table 7 extends Table 1 to include the parameter count for the models from all the sections.

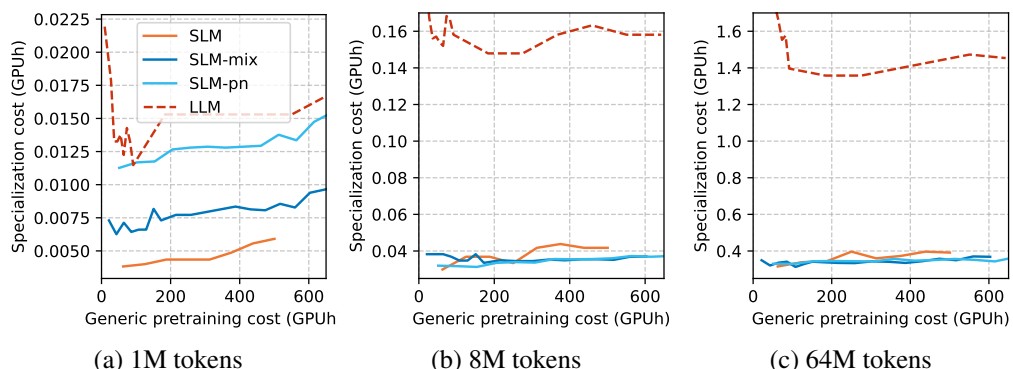

(a) 1M tokens        (b) 8M tokens        (c) 64M tokens

Figure 9: Fine tuning cost as a function of the pretraining cost.

## B  INTERPOLATED PERPLEXITIES

We report the data from the Figures 3 – 4 in Table 8. Since the methods were evaluated at a fixed frequency in steps, we linearly interpolate perplexities and step counts to report results at the same pretraining costs for all methods.

## C  NUMBER OF FINE-TUNING STEPS

Figure 9 reports the fine tuning cost each model. This cost corresponds to the number of steps to reach the best validation perplexity. It is an optimistic cost estimates as one usually needs a few more steps to assess that further improvement is not expected. The fine-tuning cost seems to grow $\sim 10X$ when the fine-tuning set size grows 8X. The LLM usually requires less steps than the SLMs but its steps are more expensive. The vanilla SLM overfits earlier than the other SLMs (SLM-mix, SLM-pn) for the small 1M specialization set but not for the larger sets.

## D  CLUSTERING

The clustering of c4 is used by the mixture model to define each expert scope. Similarly it is used as the conditioning variable by the PN. Finally it is used by importance sampling to resample c4. Table 9 reports the concentration of each specialization domain from Pile on their most frequent cluster. A high concentration could be positive since it means that, when fine-tuning SLM-pn or SLM-mix conditioned on this cluster, one starts from pretrained parameters containing most of the pretraining data relevant to the domain at hand. The table also reports the most frequent cluster on c4 to highlight that the specialization domain distributions differ from the c4 distribution.

Table 6: Transformer parameters

|  | SLM | LLM |
| --- | --- | --- |
| Architecture | | |
| Mum. layers | 7 | 7 |
| Model dimension | 1024 | 2816 |
| Inner MLP dimension | 4096 | 11264 |
| Num. attention heads | 8 | 22 |
| Optimizer | | |
| Optimizer | Adam | Adam |
| Learning rate | 1e-4 | 1e-4 |
| Clipping norm | 5.0 | 5.0 |
| Linear warmum steps | 1,000 | 1,000 |

Table 7: Number of parameters (in millions) for pretraining and inference.

| Model | | | Num. parameters (m). | |
|---|---|---|---|---|
| | | | Overall | Inference |
| SLM | | | 126 | 126 |
| SLM-pn | 16 | experts | 756 | 126 |
| | 32 | | 1,422 | 126 |
| | 64 | | 2,770 | 126 |
| SLM-mix | 4 | experts | 504 | 126 |
| | 16 | | 2,016 | 126 |
| | 64 | | 8,064 | 126 |
| | 256 | | 32,256 | 126 |
| LLM | | | 771 | 771 |

Table 8: Interpolated perplexities at fixed pretraining costs (GPUh)

| Model | Pretrain cost | Num. steps | Num. GPU | Generic PPL | Spec. PPL | | | |
|---|---|---|---|---|---|---|---|---|
| | | | | | No ft | 1M | 8M | 64M |
| SLM | 100 | 798k | 8 | 20.51 | 33.74 | 19.31 | 15.61 | 12.37 |
| SLM-mix | 100 | 464k | 16 | 17.13 | 34.35 | 19.82 | 15.82 | 12.62 |
| SLM-pn | 100 | 195k | 8 | 18.90 | 33.44 | 18.57 | 15.58 | 12.53 |
| LLM | 100 | 108k | 8 | 17.00 | 29.22 | 17.11 | 15.49 | 11.55 |
| SLM | 200 | 1597k | 8 | 19.71 | 34.43 | 18.58 | 15.12 | 12.09 |
| SLM-mix | 200 | 928k | 16 | 15.92 | 31.94 | 18.48 | 14.98 | 12.15 |
| SLM-pn | 200 | 390k | 8 | 17.74 | 32.30 | 17.76 | 14.95 | 12.13 |
| LLM | 200 | 217k | 8 | 15.58 | 28.18 | 15.62 | 14.03 | 10.81 |
| SLM | 400 | 3195k | 8 | 19.17 | 36.61 | 18.22 | 14.80 | 12.00 |
| SLM-mix | 400 | 1000k | 16 | 15.82 | 31.04 | 17.56 | 14.42 | 11.84 |
| SLM-pn | 400 | 780k | 8 | 16.90 | 32.54 | 17.17 | 14.48 | 11.86 |
| LLM | 400 | 434k | 8 | 14.54 | 28.98 | 15.03 | 13.05 | 10.28 |
| SLM-mix | 600 | 1000k | 16 | 15.82 | 31.03 | 17.18 | 14.21 | 11.73 |
| SLM-pn | 600 | 1170k | 8 | 16.53 | 32.53 | 16.95 | 14.29 | 11.74 |
| LLM | 600 | 651k | 8 | 14.09 | 28.62 | 14.50 | 12.64 | 10.07 |

## D.1 NUMBER OF CLUSTERS FOR IMPORTANCE SAMPLING

Our importance sampling strategy resamples c4 such that its cluster histogram matches the cluster histogram from the targeted domain (Pile subset). The number of clusters is an important parameter. A small number of clusters will change the c4 distribution only in a coarse manner and will provide a low fidelity match with the targeted set. Conversely, a large number of clusters has two drawbacks. Firstly, when the specialization set is small, cluster frequencies might be poorly estimated for a large number of clusters. Secondly, with a large number of clusters, the targeted histogram might concentrate a big fraction of the mass on a few small clusters, meaning that the resampled c4 dataset

Table 9: Fraction of data in the most frequent cluster, per domain.

| Domain | Num. clusters | | | | |
|---|---|---|---|---|---|
| | 4 | 16 | 64 | 256 | 1024 |
| arxiv | 0.95 | 0.92 | 0.55 | 0.52 | 0.29 |
| europarl | 0.52 | 0.53 | 0.45 | 0.44 | 0.27 |
| freelaw | 0.48 | 0.73 | 0.87 | 0.72 | 0.35 |
| gutenberg | 0.75 | 0.54 | 0.35 | 0.27 | 0.29 |
| opensubtitles | 0.97 | 0.68 | 0.26 | 0.28 | 0.32 |
| openwebtext2 | 0.53 | 0.35 | 0.12 | 0.04 | 0.02 |
| pubmed abs. | 0.94 | 0.54 | 0.41 | 0.20 | 0.06 |
| stackexchange | 0.95 | 0.94 | 0.78 | 0.61 | 0.31 |
| wikipedia | 0.71 | 0.58 | 0.21 | 0.07 | 0.03 |
| c4 | 0.32 | 0.12 | 0.04 | 0.02 | 0.00 |

will contain many repeated points from these clusters. This can degrade performance as the effective size of the resampled c4 dataset will be smaller with these repetitions.

Our main results report the importance sampling results with 1,024 clusters. Figure 10 reports the results with 16, 64, 256 and 1,024 clusters.

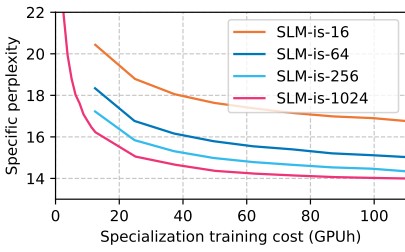

Figure 10: Specialization perplexity for importance sampling with different number of clusters after fine-tuning on 1M tokens.

## D.2 NUMBER OF CLUSTERS FOR MIXTURE OF EXPERTS

The overall size of the mixture and its training cost are proportional to the number of clusters. Our main results (Fig. 3, Fig. 4, etc) use 16 experts. We compare results with 4 to 256 experts. Intuitively, if the number of experts is too large, the model would cost more to train and each cluster would not contain enough data to train a model of the size of SLM. Conversely, if the number of experts is too small, the training cost is low but each SLM-sized expert would be trained from a large cluster and would underfit its training set. Also, the large clusters might be too generic and far from the distribution of the targeted set. Figure 11 shows the macro-averaged perplexity on the Pile as a function of the generic pretraining time for the different mixture sizes in the case of the 1M token specialization set.

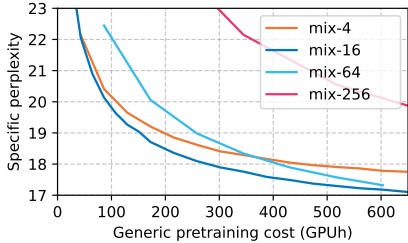

Figure 11: Specialization perplexity of mixture models with 4-256 experts on Pile subsets (average) after fine-tuning on 1M tokens.

## E INDIVIDUAL SUBSET RESULTS

Figure 12 decomposes the results in Figure 4 (b) per domain. The subset results are mostly consistent with the average but we observe few differences. SLM-pn and SLM-mix have a close average and the best method among them varies per subset. Also we notice that both methods do not outperform SLM on wikipedia and openwebtext2. The disadvantage of SLM-pn and SLM-mix over SLM can be observed before fine-tuning, as shown on Figure 13. We report the entropy of the cluster histograms in Table 10 and observe that wikipedia and openwebtext2 are the domains with the highest entropy. This means that the c4 data similar to these datasets is more spread across clusters than for the other domains. Conditioning SLM-pn and SLM-mix on a single cluster variable might not model well these domains. Of course, this correlation between entropy and fine-tuned perplexity of SLM-mix, SLM-pn could be fortuitous. This motivates us to investigate the impact of the different clustering methods and their metrics in future research.

Table 10: Entropy of the cluster histogram for each domain.

| Domain | Num. clusters | | | |
|---|---|---|---|---|
| | 16 | 64 | 256 | 1024 |
| arxiv | 0.41 | 1.02 | 1.80 | 2.58 |
| europarl | 1.48 | 1.83 | 2.31 | 3.14 |
| freelaw | 1.01 | 0.70 | 1.44 | 2.49 |
| gutenberg | 1.57 | 2.42 | 3.21 | 3.85 |
| opensubtitles | 1.16 | 2.61 | 2.95 | 3.44 |
| openwebtext2 | 2.19 | 3.60 | 4.89 | 6.12 |
| pubmed abs. | 1.07 | 2.14 | 3.22 | 4.43 |
| stackexchange | 0.39 | 0.97 | 1.78 | 3.24 |
| wikipedia | 1.73 | 3.20 | 4.54 | 5.64 |
| c4 | 2.73 | 4.07 | 5.46 | 6.85 |

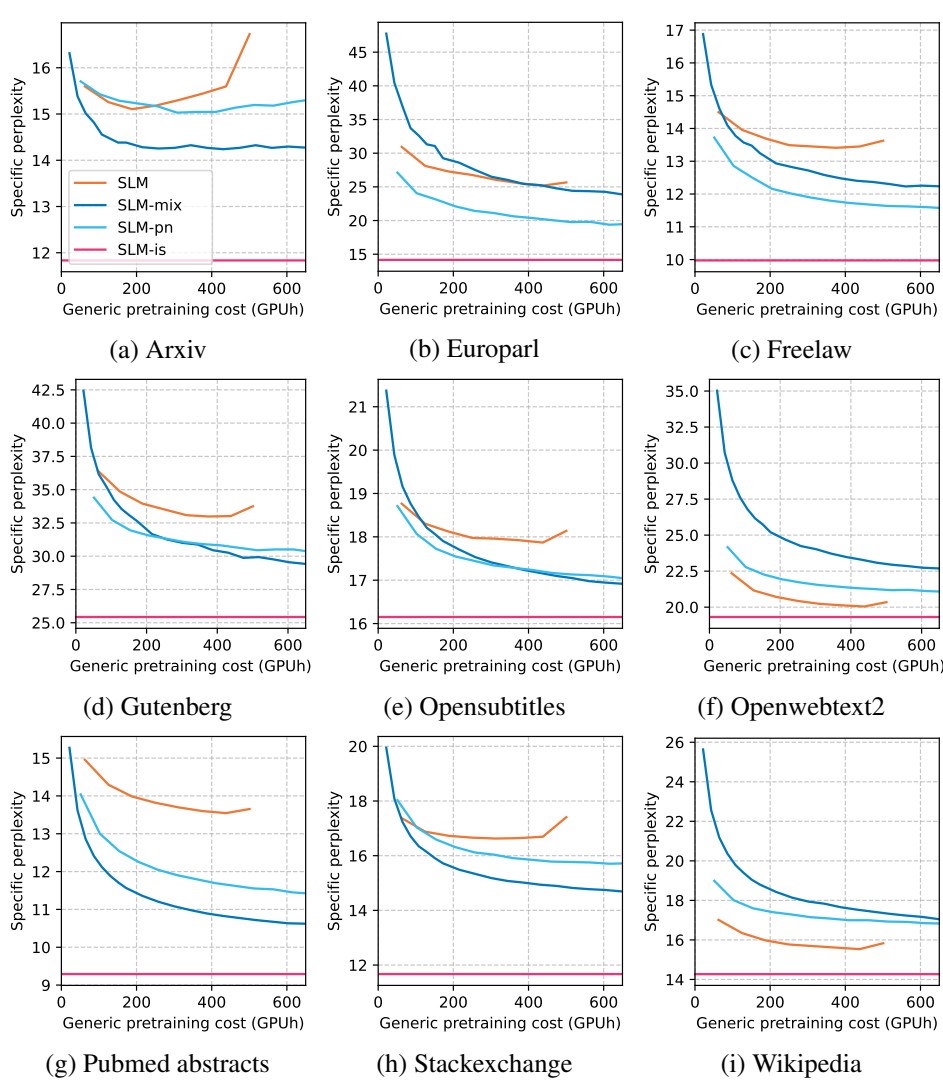

Figure 12: Specialized perplexity on individual subsets after fine-tuning on 1M tokens.

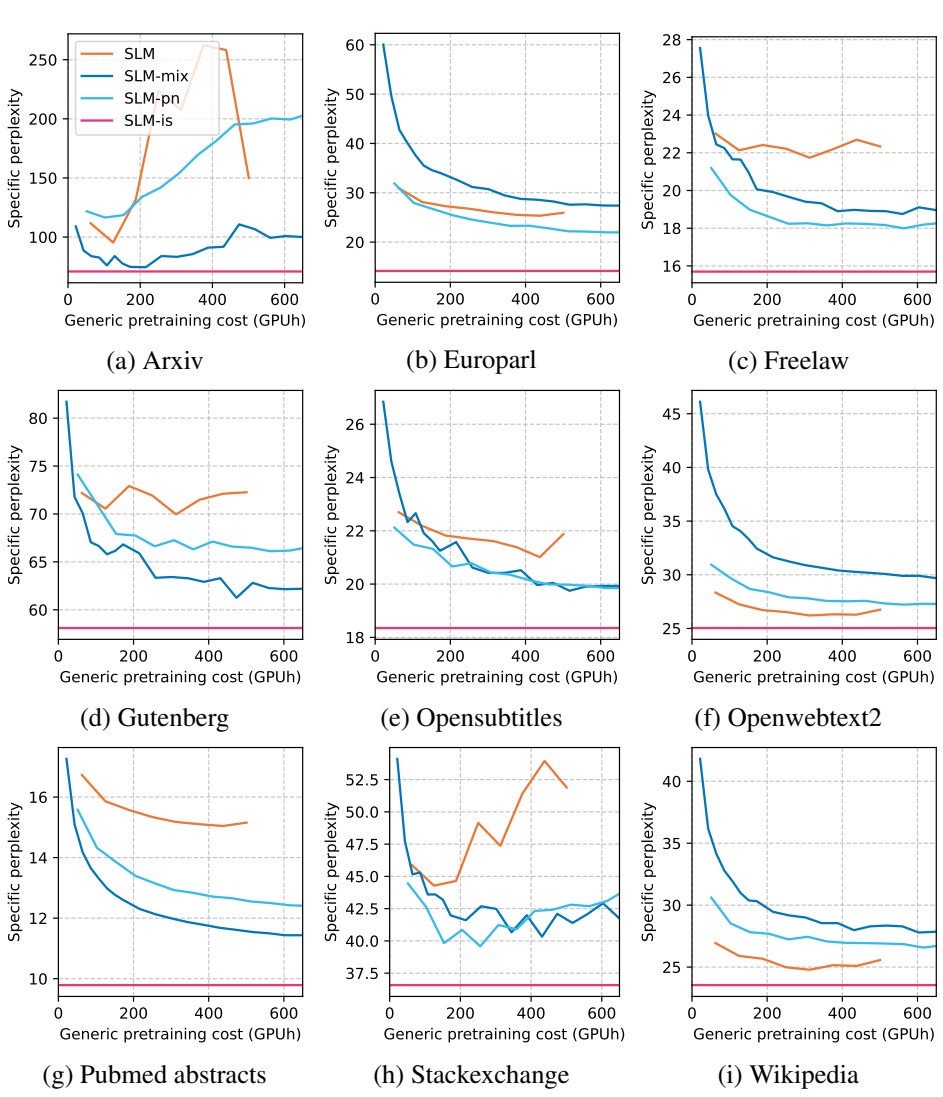

Figure 13: Specialized perplexity on individual subsets before fine-tuning on 1M tokens.

# F  PARAMETER EFFICIENT FINE-TUNING

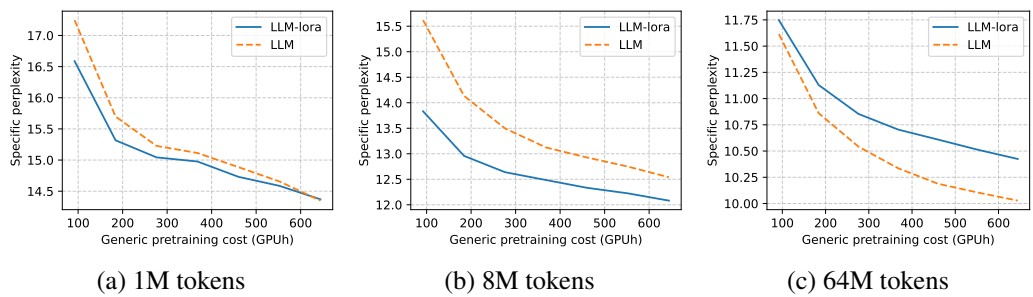

(a) 1M tokens          (b) 8M tokens          (c) 64M tokens

Figure 14: Specialized perplexity of LoRA fine-tuning on the Pile subsets with respect to the pretraining cost. We observe that LoRA fine-tuning performs very similarly to traditional fine-tuning with less than 0.5 perplexity differences.

We also evaluate Low Rank Adaptation (LoRA) Hu et al. (2021) as a fine-tuning method for the LLM. LoRA can help regularize the fine-tuning process when little specialization is available. It also reduces the storage and communication costs of managing many specialized models when addressing many domains since only few parameters are learned for each domain. LoRA does not reduce the pretraining cost, and even increases the fine-tuning cost as it requires more fine-tuning steps, with a similar cost per step. In our LoRA experiments we use low-rank matrices of rank 64 which results in 5M trainable parameters and fine-tune for up to $5\times$ more steps than for the LLM. We observe that LLM-lora required from $25\%$ more steps than the LLM for the 1M token dataset and $3\times$ more steps for the 64M token dataset. However, since the specialization cost is negligible in comparison to the pretraining cost these extra steps do not really impact the overall training cost. Figure 14 reports the results. LoRA performs very similarly to the LLM (differences of less than $0.5$ perplexity) and with the exception of the "large" domain-specific regime of 64M tokens we can observe some ovefitting mitigation. Finally, LoRA still results in a large model which is not suitable for the cases where the computational budget for inference is small.

