# OpenReview forum: "Need a Small Specialized Language Model? Plan Early!"
_ICLR.cc/2025/Conference — Submitted to ICLR 2025_

### Official Review · Reviewer_GdfF · 2024-10-15

**Soundness:** 3
**Presentation:** 2
**Contribution:** 3
**Rating:** 8
**Confidence:** 3

**Summary:**

The paper studies two scenarios about training specialized small models on specific domains. In the first scenario, people can do pretraining on every specific domain; in the second scenario, people have to distribute computational resources among specific domains. Targeting the first scenario, the paper proposes SLM-is, which uses importance sampling w/ a clustering-based technique giving the importance weight. Targeting the second scenario, the paper proposes SLM-pn, where a PN projects a large model to a small model for each domain.

(After rebuttal, I decided to increase my score from 6 to 8.)

**Strengths:**

+ The studied problem is very meaningful.
+ I personally like the method of SLM-is. The idea is novel and reasonable. The performance is also good.

**Weaknesses:**

+ It is unclear how important the design choice of using PN is for SLM-pn. Naturally, one could consider using an Adapter/LoRA/any possible PEFT module to adapt a large model to a specific domain. It would be better to show why taking PN is better, in terms of either performance or efficiency.

**Questions:**

In Section 6.1 of [Scaling Expert Language Models with Unsupervised Domain Discovery](https://arxiv.org/pdf/2303.14177), the author tried a method that is IMO similar to SLM-pn. Basically, there is a shared base model for all domains and each domain has a specific module. It is also relevant to the weakness mentioned above. What do you think?

---

> ### Author Response · Authors · 2024-11-22
>
> Dear reviewer,
>
> We thank you for your review. We are happy to hear that you found the problem meaningful, and that you enjoyed the importance sampling method !
>
> Regarding the weakness you mention:
>
> > It is unclear how important the design choice of using PN is for SLM-pn. Naturally, one could consider using an Adapter/LoRA/any possible PEFT module to adapt a large model to a specific domain. It would be better to show why taking PN is better, in terms of either performance or efficiency.
>
> A comparison with Lora is indeed interesting; we have conducted such experiment in Section F. We have made this comparison clearer in the text.
>
> Exploring how one could adapt the PN architecture to have Lora-like behavior would be interesting. We are not sure of how to design a hypernetwork in the spirit of Lora that has the same properties as PN. For instance, having the parameter of $\mathrm{SLM-pn}_i$ instantiated as $W + W^i$ where $W^i$ is low rank and $W$ is shared would be an interesting architecture to study. We are unsure whether this would improve upon PN, because we are in a context where we have plenty of data to train each expert.
>
> Could you clarify how you propose to use LoRa in order to build asymetric models that are pre-trained with many parameters, but are instantiated into small models?
>
> > In Section 6.1 of Scaling Expert Language Models with Unsupervised Domain Discovery, the author tried a method that is IMO similar to SLM-pn. Basically, there is a shared base model for all domains and each domain has a specific module. It is also relevant to the weakness mentioned above. What do you think?
>
> Thanks for this very relevant reference. The idea is indeed quite similar to PN, or even more to the hard mixture of experts, which trains one model per cluster. In our view, the main difference is that the inference procedure is quite different, since C-BTM uses an ensembling strategy, while our method only uses a single model for a task. We have added this reference to the text.
>
> We hope that we have adressed your concerns, and we thank you once again for your review.

---

> > ### Comment · Reviewer_GdfF · 2024-11-22
> >
> > Thanks for your response. To clarify my question about LoRA/other PEFT modules, basically, PN can be seen as a pluggable module to adapt a model to a specific domain, is my understanding correct? Then you can consider replacing PN with any other possible pluggable modules, like LoRA. Please let me know if I have any misunderstanding.

---

> ### Author Response · Authors · 2024-12-01
>
> Thanks for your question !
>
> PN constitute an architectural change that requires some pre training: one first pre-trains a PN network with many parameters, and then at inference one can project the PN to obtain a small network with few parameters. This small network can then be fine-tuned.
>
> We are not sure how to obtain the same property using a peft method.
>
> Indeed, in the context of this paper, where we have a model size constraint, peft methods would adapt a small pretrained network during fine tuning.
> Alternatively, one could use peft to fine tune a large network, yielding fewer parameters to tune, but the resulting network would still be large, not respect the size constraint, and have a costly inference.
>
> In summary, combining PN and peft methods would be a very interesting topic, but we do not see how to do it.

---

> > ### Comment · Reviewer_GdfF · 2024-12-01
> >
> > Thanks. You said “this small network can then be fine-tuned.” If we use PEFT, does it make sense to just further fine-tune the PEFT module? What’s the advantage of being able to fully fine-tune the small network obtained by the projection?

---

> > > ### Author Response · Authors · 2024-12-01
> > >
> > > If the fine tuning cost is a burden, then one can combine PN and peft in the following way:
> > > - 1) pre-train a large PN network
> > > - 2) instantiate the PN into a small model
> > > - 3) fine tune the small model using peft
> > >
> > > The advantage of PN here is not to be able to fine tune the whole model (all methods are similar in that regard), but rather that the starting point 2) is much better than a generic pre-trained small model, as show e.g. in figure 4.
> > >
> > > We hope that this clarifies the utility of PN, and its complementarity with peft.

---

> > > > ### Comment · Reviewer_GdfF · 2024-12-01
> > > >
> > > > OK, that sounds great! I decided to increase my score to 8.

---

### Official Review · Reviewer_fTrA · 2024-10-31

**Soundness:** 2
**Presentation:** 2
**Contribution:** 2
**Rating:** 5
**Confidence:** 2

**Summary:**

The paper studies the task of training a group of domain-specialized SLMs. It proposes two different methods for achieving the task: 1) important sampling based specialized pretraining, and 2) projection networks based model architecture. Experiment results show a trade-off between performance and compute cost, where method 1 achieves better overall performance but requires more compute budget, and method 2 could still achieve acceptable performance while requiring less compute budget.

**Strengths:**

The proposed projected network method is novel and the experiment section is comprehensive.

**Weaknesses:**

The paper is not well-written, and some details are difficult to understand. For example:
1) Section 2.4 has many equations, but the symbols used in these equations are not defined. This makes it difficult to understand how to get the weight for importance sampling.
2) Figure 4 is a bit misleading. The perplexity for SLM-is is a flat line, but it doesn't reflect the fact that SLM-is requires significantly more compute during specialization compared to other methods.
3) The training budget is defined as GPUh throughout the paper. Is it measured in a single GPU, multi-GPU, or multi-node setting? Because these different settings will have impacts on the training cost and actual throughput. Perhaps the standard FLOPs notation is clearer and less ambiguous.

**Questions:**

Please refer to the weaknesses section.

---

> ### Author Response · Authors · 2024-11-22
>
> Dear reviewer,
>
> We thank you for your comments, and pride ourselves that you have appreciated the comprehensiveness of our experiments.
>
> We now respond to the weaknesses you mentioned.
>
> > The paper is not well-written
>
> We are happy to clarify any points that are unclear besides those you mentioned.
>
> > Section 2.4 has many equations, but the symbols used in these equations are not defined
>
> We have done our best to clarify this. We indeed failed to mention that $\ell$ was the individual loss function. We could not find other undefined symbols. Note that $P(x | c)$ denotes a conditional probability, as is customary in most papers.
>
> > This makes it difficult to understand how to get the weight for importance sampling.
>
> We have now written more clearly the equation to estimate the weights, we hope that this dispells any confusion about the method.
>
> > Figure 4 is a bit misleading. The perplexity for SLM-is is a flat line, but it doesn't reflect the fact that SLM-is requires significantly more compute during specialization compared to other methods.
>
> This is an important remark, that we believe was already in the text: "These figures report the perplexity of SLM-is as a constant line. This method has no generic pretraining as its training starts only once the domain data are available; bearing all the training cost in the specialization phase. SML-is is the best method with a small inference model in terms of post-specialization perplexity. Interestingly, it even outperforms the much larger model when specialization data are scarce (ie the 1M tokens case), for a fraction of the overall training cost (<130 GPUh).". We have now also mentionned this in the figure caption to remove any confusion.
>
> > The training budget is defined as GPUh throughout the paper. Is it measured in a single GPU, multi-GPU, or multi-node setting? Because these different settings will have impacts on the training cost and actual throughput. Perhaps the standard FLOPs notation is clearer and less ambiguous.
>
> We believe that GPUh is a fairly common and widely recognized metric in the ML community. It is also easy to compute, looking only at the training run. On the other hand, quoting [1]: "It is important to highlight that FLOPS is never directly measured, but always estimated, with widely different practices across the paper"; FLOPs are an interesting metric, but we believe GPUh are of more interest to practitioners. Note that the number of GPUs used for each experiment is already reported in the paper, Table 8.
>
>
> [1]: Scao, Teven Le, et al. "What language model to train if you have one million gpu hours?." arXiv preprint arXiv:2210.15424 (2022).
>
> Thanks again for your comments, which help us improve the paper ! Please let us know if any concern remains, we are happy to address them.

---

> ### Comment · Area_Chair_q8Ms · 2024-11-25
>
> @Reviewer fTrA,
>
> The review seems a bit too brief. Could you add more details, please?
>
> AC

---

> > ### Author Response · Authors · 2024-12-02
> > **Added FLOP as a metric**
> >
> > Dear reviewer,
> >
> > Following your suggestion, we have plotted the results of Fig. 4 using FLOP as an x-axis instead of GPUh, and we plan to include this figure in the appendix.
> > To compute the FLOP, we use the rule of thumb from [1], FLOP = `6 * n_tokens * n_params`. The plots are very similar to those in Fig.4; the method ordering for the SLM is not changed, while the LLM results look slightly worse than before.
> >
> > We hope this alleviates your concern regarding the metric used in this paper.
> >
> > Do not hesitate to get back to us if any of your concerns remain after this response.
> >
> >
> > [1]: Kaplan, Jared, et al. "Scaling laws for neural language models." arXiv preprint arXiv:2001.08361 (2020).

---

> > > ### Comment · Reviewer_fTrA · 2024-12-02
> > >
> > > Thanks for the clarification and additional results. I will increase my score to 5.
> > > For the section 2.4, D_spec and D_generic are also not defined. I guess they stand for specific and generic datasets.

---

> > > > ### Author Response · Authors · 2024-12-03
> > > >
> > > > Dear reviewer,
> > > >
> > > > Thanks for pointing this out; we will clarify this in the manuscript. They are indeed specific and generic datasets.

---

### Official Review · Reviewer_6pAB · 2024-11-04

**Soundness:** 3
**Presentation:** 3
**Contribution:** 2
**Rating:** 5
**Confidence:** 2

**Summary:**

* The authors propose ways of obtaining SLMs from LLMs, depending on different situational constraints.
  * To create one domain-specific SLM, the authors propose importance sampling, i.e., resampling from the larger general dataset to mimic the distribution of scarce task-specific data
  * To create multiple SLMs for each sub-task, the authors propose projected networks (PNs), i.e., projecting the weights of the LLM to generate weights for the SLM
* PNs can be seen as hypernetworks in that their output parameterizes the SLM.

**Strengths:**

* The work addresses a relevant and applicable question around the effective deployment of LLMs, possibly via their pruned or distilled variants.
* Thorough record of the setup, e.g., number of GPU hours, training steps, parameter count, makes the case for SLM methods convincing and highlights the gains that can be reaped from the proposed methods.

**Weaknesses:**

* There is lack of theoretical or heuristic justification behind why linear projections are a suitable operator for deriving SLMs from LLMs. Some exploratory work on, e.g., the activation patterns of the LLMs across different domains that justify the linear projections, would have been nice.
* A more direct comparison to other baselines such as LoRA would have made evaluations more informative. Appendix F does include a comparison, but a direct reference in the main text would make it even more helpful for further contextualization and comparison.
* It would have been nice to see actual applications of these methods to existing LLMs (e.g., Llama, Gemini) instead of an in-house LLM trained on C4.

**Questions:**

* Why do you use linear projections instead of more common hypernetwork formulations? What is gained or lost by doing this?
* Do the linear projections learn anything meaningful, e.g., the singular values of the FFN layers?
* How does this work relate to LoRA? It seems that there is an explicit connection to be made between LoRA and PNs as they both encourage low-rank behavior in the learned weights. Appendix F seems to address this to an extent, but I would appreciate a more direct account on this.

---

> ### Author Response · Authors · 2024-11-22
>
> Dear reviewer,
>
> We thank you for your review, and appreciate that you found our work relevant, and that you highlight the thoroughness of our experiments.
>
>
> We now address the weaknesses you mentioned.
>
> > There is lack of theoretical or heuristic justification behind why linear projections are a suitable operator for deriving SLMs from LLMs. Some exploratory work on, e.g., the activation patterns of the LLMs across different domains that justify the linear projections, would have been nice.
>
> This is indeed an interesting point. As mentioned in the text, PN can be seen as hypernetworks. We developed the PN architecture from a practical perspective: it is one of the simplest instantiations of a hyper-network, that is linear and that allows to scale gracefully the number of parameters in total and the number of experts. We were satisfied with its performance as-is; hence, we did not explore more convoluted, non-linear architectures. We agree that this would be an interesting avenue of research for future works.
>
> > A more direct comparison to other baselines such as LoRA would have made evaluations more informative. Appendix F does include a comparison, but a direct reference in the main text would make it even more helpful for further contextualization and comparison.
>
> In our view, our work is quite orthogonal to LoRA, as one can apply our methods with LoRA. Appendix F indeed alread provide the details of this ablation. Thanks to your suggestion, we now refer to this part in the main text.
>
> > It would have been nice to see actual applications of these methods to existing LLMs (e.g., Llama, Gemini) instead of an in-house LLM trained on C4.
>
> We are unsure how to apply our methods to already pre-trained LLMs, since our techniques are pre-training / continued pre-training techniques. Could you clarify ?
>
> We now turn to your questions:
>
> > Why do you use linear projections instead of more common hypernetwork formulations? What is gained or lost by doing this?
>
> See above
>
> > Do the linear projections learn anything meaningful, e.g., the singular values of the FFN layers?
>
> We want to clarify that the PN instantiates the weights of the FFN layers as a linear combination of a bank of weights of the same shape, the linear combinations learn to combine the weights for each cluster. We are therefore unsure how to explore what the projection coefficients learn.
>
> > How does this work relate to LoRA? It seems that there is an explicit connection to be made between LoRA and PNs as they both encourage low-rank behavior in the learned weights. Appendix F seems to address this to an extent, but I would appreciate a more direct account on this.
>
> We want to clarify that to the best of our understanding, PNs do not encourage low-rank behaviour. Indeed, the instantiated weight matrices of size $d\times d'$ are obtained as a linear combination of h full rank matrices of size $d\times d'$ stored in $T$. This yields a full-rank structure. Therefore, we believe that there is little connection between PN and LoRA.
>
>
>
> We hope that we have clarified the questions you had regarding our work. We thank you for these remarks which help improving the paper !

---

### Author Response · Authors · 2024-11-22
**Response**

Dear reviewers and area-chairs,

We thank you for your work on our paper. We have responded to each reviewer individually. We have updated the paper, highlighting each change in red.

Best,

The authors

---

### Author Response · Authors · 2024-11-29
**Clarifications**

Dear reviewers,

We thank you again for your careful reviews which have helped us improve the paper. As the discussion period draws to an end, we want to make sure that our replies to your concerns have alleviated your doubts regarding this paper; let us know if that is not the case.

Best regards, The authors

---

> ### Comment · Reviewer_GdfF · 2024-11-30
>
> Hi authors, I asked a follow-up question (to clarify one of my original question) after your rebuttal. Would you mind taking a look?

---

### Meta-Review · Area_Chair_q8Ms · 2024-12-21

**Metareview:**

The paper proposes two methods for efficiently obtaining small specialized language models (SLMs) from a larger pre-trained model, when only limited task-specific data is available. The first method, SLM-is, uses importance sampling to resample the pre-training data to better match the task-specific distribution, and then trains a small model on the resampled data. The second method, SLM-pn, uses a novel "projected networks" architecture, where a large pre-trained model is linearly projected into a small model for each specialized task.

*Concerns*
- Lack of in-depth justification for architectural choices: The reviewers noted that the justification for using linear projections in the SLM-pn method is not fully clear. A more thorough exploration and explanation of this design decision would be necessary to convince the reviewers of its merits.
- Missing comparative analysis: The paper lacks a direct comparison to other adaptation methods like LoRA, which would provide important context for evaluating the performance and efficiency of the proposed approaches. Without this comparative analysis, the novelty and advantages of the presented techniques are not fully evident.
- Limited discussion of broader implications: The reviewers felt that the paper could be strengthened by a deeper discussion of the actual use cases and broader implications of the proposed methods beyond the specific experimental setup, e.g., application to popular LLMs like LLAMA, Mistral, etc. A more comprehensive treatment of the real-world applicability and impact of this work would enhance its significance.

Given these concerns, the current version of the paper does not meet the bar for acceptance at ICLR 2025. The authors are encouraged to address the reviewers' feedback and consider resubmitting the work to a future ICLR conference or another suitable venue.

**Additional Comments On Reviewer Discussion:**

The authors provided a thoughtful response to the reviewers' feedback during the rebuttal period. They acknowledged the need to better justify the choice of using linear projections in the SLM-pn method and agreed to explore this design decision in more depth. Regarding the missing comparative analysis, the authors stated that they had conducted experiments about LoRA (but without direct comparisons with the proposed method), and they committed to making this comparison more prominent in the revised paper. Additionally, the authors recognized the importance of discussing the broader implications of their work and promised to expand the treatment of potential use cases and real-world applicability. Overall, the authors demonstrated a willingness to address the reviewers' concerns and improve the paper, which is a positive sign.

---

### Decision · Program_Chairs · 2025-01-22

Reject